# Germination and Growth Characteristics of *Quercus myrsinifolia* Blume Seedlings According to Seed Coat Removal, Type of Potting Soil and Irrigation Cycle

Eun-Ji Choi [1], Seong-Hyeon Yong [2], Dong-Jin Park [3], Kwan-Been Park [2], Do-Hyun Kim [2], Eon-Ju Jin [4] and Myung-Suk Choi [2,*]

1   Plant Conservation Division, Korea National Arboretum of the Korea Forest Service, Pochen 11186, Korea; ch264000@naver.com
2   Division of Forest Environmental Resources, Institute of Agriculture and Life Science, Gyeongsang National University, Jinju 52828, Korea; ysh1820@naver.com (S.-H.Y.); qls4347@naver.com (K.-B.P.); danny9605@naver.com (D.-H.K.)
3   Department of Seed and Seedling Management, National Forest Seed and Cultivar Center, Chungju 27495, Korea; djp0903@korea.kr
4   Forest Biomaterials Research Center, National Institute of Forest Science, Jinju 52828, Korea; jinej85@korea.kr
*   Correspondence: mschoi@gnu.ac.kr; Tel.: +82-772-1856

**Abstract:** The importance of evergreen oak species is increasing due to changes in the ecosystem caused by climate change and environmental changes such as fine dust and carbon dioxide. The *Quercus myrsinifolia* Blume seeds showed a recalcitrant seed property, where the germination rate decreased when the moisture content was decreased. For seedling propagation of evergreen oak, the effect of oak seed coat (pericarp and testa) removal on germination and seedling growth as well as the effect of potting soil and irrigation cycle on seedling quality were investigated. The germination rate and germination characteristics of *Q. myrsinifolia* evergreen oak seeds showed significant differences depending on the storage period and the presence or absence of seed coat. Seed coat removal significantly increased germination rate compared to intact seeds, accelerated mean germination time, and increased germination rate and germination value. There was no significant difference in germination rate according to the storage period. The growth of *Q. myrsinifolia* seedlings was much better in the seeds with the seed coat removed than the intact seeds. The root collar diameter of seedlings germinated from intact seeds was 2.44 mm, and the root collar diameter of seedlings from which the seed coat was removed was 3.38 mm. As a result of the growth characteristics according to the potting soil, 1- and 3-year-old *Q. myrsinifolia* seedlings showed excellent root growth in commercial potting soil and sand mixed potting soil. Consequently, seedling quality index was 0.124–0.257 according to irrigation and 0.149–0.262 according to potting soil. From observing the root growth of the seedlings according to the irrigation treatment, in the case of 3-year-old seedlings, the total root length was 432 cm when irrigated every 3 days, and the growth was the best. The above results are expected to contribute significantly to the mass propagation of *Q. myrsinifolia*, which is important for warming and urban greening.

**Keywords:** evergreen oak; irrigation; *Quercus myrsinifolia* Blume; recalcitrant seed; seed coat removal; potting soil; seedling quality index; seed storage





## 1. Introduction

The growth of plant distribution includes competition among populations, invasion by alien species, changes in topography, and human intervention. Among them, climate change is known as the most significant factor affecting changes in vegetation zones. In Korea, as the temperature rises, pine trees' vegetation zone is decreasing, and the power of temperate evergreen broadleaf trees is expanding to the north [1,2].

Optimized seedling techniques are critical as temperate evergreen hardwood resources are becoming increasingly important. The success of these seedlings depends on seedling technique, genetic factors, seedling environment, and seeding timing and technique [3]. Bare root seedlings produced in nurseries have mainly been used for planting oak species in Korea, but container seedlings produced in facilities have recently been planted a lot. Oak tree species develop rectilinear roots and, thus, have excellent main root development but relatively low development of lateral roots and fine roots [4]. However, there are few studies on the proliferation of *Quercus myrsinifolia* Blume.

In the container nursery system, control of the growth environment, such as through light, temperature, moisture, the type of container, the potting soil, and the fertilization technique, has a significant influence [5,6]. In particular, moisture significantly affects tree growth, photosynthetic activity, fertilizer efficiency, nutrient availability, and stress due to a lack of moisture, which leads to poor seedling production [7]. Moisture changes the temperature conditions in the greenhouse during the facility nursery process. It affects the respiration and transpiration of container seedlings, resulting in changes in plant physiology mechanisms such as photosynthetic capacity, stomatal conductivity, and water efficiency [8–10]. It also affects root growth, short root, and root rotation of container seedlings by changing the aeration and moisture content in the growing potting soil [11]. Moisture is one of the most important inorganic environmental factors in the planting process. A study on the irrigation cycle and irrigation amount, which are most closely related to moisture, should be conducted.

Evergreen *Quercus* trees are promising tree species in preparation of climate change due to their tolerant ability to heat and drought–heat dual stress [12]. In the Korea peninsula, there are six species of evergreen *Quercus* species: *Quercus acuta* Thunb., *Quercus gilva* Blume, *Quercus glauca* Thunb., *Quercus myrsinaefolia* Blume, *Quercus phillyraeoides* A. Gray and *Quercus salicina* Blume. These typical evergreen tree species of the warm-temperature zone naturally grow in the south shore or southern islands of the Korean peninsula.

*Q. myrsinifolia* is an evergreen tree that grows on the coasts of islands in Jeonlanam-do and Gyeongsangnam-do in Korea, in valleys below 700 m above sea level, at the foot of mountains and on Jeju Island throughout the temperate and tropics of the Northern Hemisphere. *Q. myrsinifolia* are used as windbreaks, fire protection, and hedges. In addition, the tree trunk of the *Q. myrsinifolia* is hard and strong, so its uses are diverse, such as textile machinery, industrial tools, ship materials, building materials, and handicrafts, and the fruits are used as food for various economic tree species. The leaves of *Q. myrsinifolia* are dense and glossy, have a beautiful appearance and strong durability, and are known to be a very promising tree for carbon absorption and particulate matter absorption [13]. Propagation of *Q. myrsinifolia* is not suitable using cuttings, and it is difficult to produce seeds because of the long distance, so efficient propagation technology development is required [14].

Container nurseries of evergreen oaks are known to be time-consuming and labor-intensive. In particular, the species of this study, *Q. myrsinifolia*, is a recalcitrant seed known to be very difficult to germinate. In addition to promoting seed germination, recalcitrant species are important for nurturing efficient seedlings. In this respect, germination and subsequent growth with or without an acorn seed coat may also be affected. In addition, the nursery technology of pot seedlings of woody plants is also very important in terms of supplying excellent seedlings. As for the growth of germinated plants, conditions such as irrigation and potting soil will significantly affect the efficiency of the nursery. This study investigated the effect on seed germination and seedling growth, potting soil type and irrigation cycle according to seed coat removal for efficient propagation of *Q. myrsinifolia*, a promising afforestation tree species for climate change.

## 2. Materials and Methods

### 2.1. Plant Materials

The seeds used in this study were collected from a 25-year-old healthy *Q. myrsinifolia* growing in Gyeongsang National University and Gajwa Experimental Forest for 3 years from September 2017 to November 2019 (Figure 1). As for the seeds, 30-year-old healthy and well-fruited trees were selected over 3 years and used for experiments. Identification of *Q. myrsinifolia* plants and seeds was carried out with the advice of Professor Hyunsik Moon, Department of Forestry Environment Resources, Gyeongsang National University. All experiments, such as moisture content and germination rate tests, germination efficiency and growth tests according to storage period, were performed using the seeds of this one tree to reduce the differences between individuals.

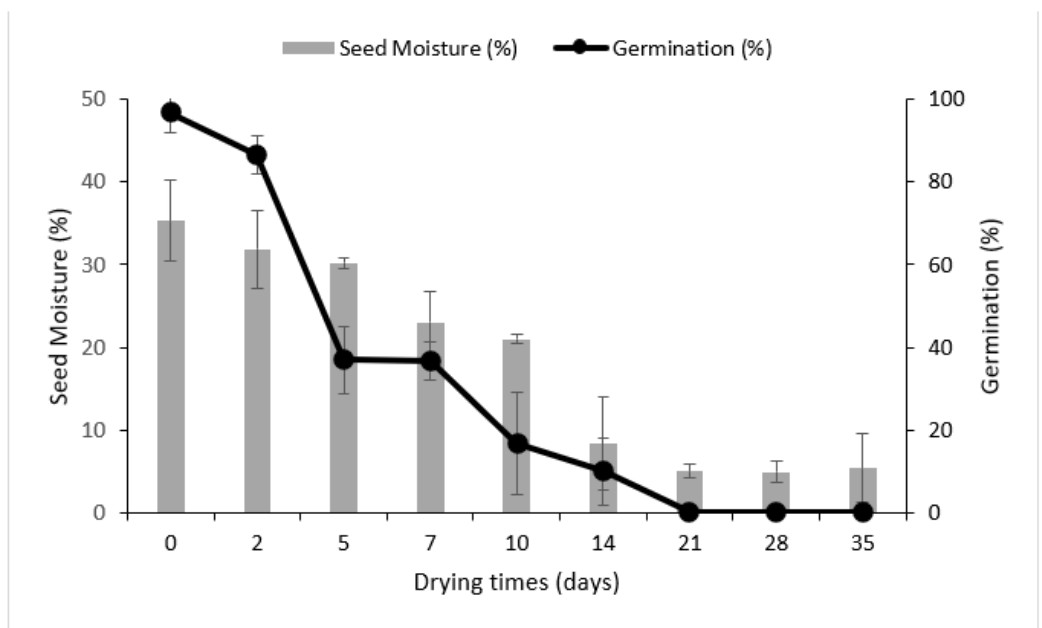

**Figure 1.** The seed moisture and germination of *Q. myrsinifolia* according to drying times. After the seeds were collected, they were immediately placed in paper bags and stored at room temperature (25 °C), and the seed moisture content and germination rate were investigated.

When the acorns turned pale yellow, they were harvested from the tree. Afterward, acorns with cupules were dried under a well-ventilated shade at room temperature for 7 days, and after removing the cupules, only healthy acorns were stored in a refrigerator at 4 °C for 1 month. The selected seeds were placed in a plastic bag soaked in water to maintain humidity, sealed, and stored in a refrigerator at 4 °C until the germination test.

For sowing, the acorns were first removed using scissors and a knife to avoid damage to the embryo and then immersed in sterile water for 12 h to remove the pericarp. In this experiment, the removal of the seed coat means that both the pericarp and testa of the acorns were removed. In this study, the seed coat refers to pericarp and testa, but from now on, it is expressed as a seed coat.

The size of the plastic container used for the germination was 420 × 280 × 140 (L × W × D, mm), the cavity volume was 320 mL, the cavity diameter was Ⓓ64 at the top and Ⓓ42 at the bottom, and the number of cavities per container was 24. For growth according to the storage period of the seeds, 1-year stored seeds and 3-year stored seeds were used.

Three-year-old *Q. myrsinifolia* seeds were sown in March 2017, and 1-year-old *Q. myrsinifolia* seeds were used in March 2019. Each seed, stored for 3 years and 1 year, was tested for germination on the same day. The germinated seedlings were then used for experiments without transplanting them into other pots. The seedlings used in the experiments to observe the irrigation cycle and the growth of seedlings planted in the potting soil

were used after transplanting 1-year-old and 3-year-old plants into 24 pots and stabilizing them for a month. No fertilization was performed during this experimental period.

### 2.2. Determination of the Relationship between Seed Moisture Content and Germination Rate

To investigate the correlation between seed moisture content and germination rate, seeds were collected from 30-year-old healthy *Q. myrsinifolia* growing on the campus of Gyeongsang National University. Seed collection was carried out on 20 November, and the germination rate was investigated according to the drying period by putting them in a paper bag at room temperature of 25 °C. Seed moisture content and germination rate were investigated 2, 5, 7, 10, 14, 21, 28, and 35 days after seed collection. The moisture content of *Q. myrsinifolia* was calculated by the following formula.

$$\text{Moisture Content } (\%) = \frac{\text{Weight of fresh seeds } - \text{ Weight of dry seeds}}{\text{Weight of fresh seeds}}$$

The germination rate of the seeds was investigated by the method of the International Seed Testing Association [15] in the seeds preserved under dark conditions for 1 month in a 25 °C incubator.

### 2.3. Germination and Seedling Growth Characteristics of Seeds with and without Seed Coat

To investigate the effect on germination and seedling growth, germination experiments were performed by selecting healthy seeds. The container where the seeds were sown was placed in a growth control room controlled at 25 °C, 2000 lux light, and photoperiod (light/dark: 16/8 h), and commercially available potting soil (CPS, Bio potting soil, Heungnongmyo, Pyeongtaek, Korea) was placed in the container. Seed germination rate and seedling growth with and without seed coats were examined. The seed coat was extracted using a scalpel. After, the seeds with and without the seed coat removed were sown in 24-cavity (0.32 L/cavity) containers with potting soils. This condition is recommended by the International Seed Testing Association [15], and irrigation is performed at 10 L once every 3 days to maintain moisture.

Seeds were considered to be germinated if the seed had more than 5 mm of cotyledon after sowing. The seedling growth indicators of *Q. myrsinifolia* included root diameter, shoot length, root length, number of leaves, and leaf length. Germination Percent (GP), Germination Rate (GR), Mean Germination Time (MGT), and Germination Value (GV) of *Q. myrsinifolia* Blume were investigated by counting the number of germinated seeds every 24 h after sowing. The following formula calculated the indicators in the 24 cavities (Table 1).

**Table 1.** Formulas for measuring germination and growth of selected *Q. myrsinifolia* seeds.

| |
| --- |
| Germination Percent (GP) = (N/S) × 100 |
| Germination Rate (GR) = $\sum(n/t)$ |
| Mean Germination Time (MGT) = $\sum(n_i t_i)/N$ |
| Germination Value (GV) = PV × MDG |

(N: total number of seed germinated; S: number of seeds planted; $t_i$: number of days after investigation time; $n_i$: germination number on the day of investigation; N, $\sum n_i$: total number of seed germinated; PV (peak value): cumulative germination rate/peak number of days after investigation time; MDG: mean daily germination).

### 2.4. Growth Characteristics of Seedlings by Irrigation and Potting Soil Types

The culture potting soil, CPS and sand mixed soil (CPS: sand 1:1, *v/v*), was used. The CPS used in this study was a mixture of 20% coco peat, 59.26% peat moss, 20% perlite, 0.632% dolomite, 0.0008% wetting agent, and 0.1% fertilizer. These soils were autoclaved for 121 °C and 30 min before being put in the pot.

Irrigation experiments were conducted for 5 months. Irrigation was performed at 1, 3, and 7 days intervals. Twenty-four seedlings were used for each treatment, and 10 L of water was irrigated. Seedling growth was measured by an indicator of height and root diameter.

To measure the growth, the potting soil was removed by carefully washing the plants separated from the pots with water so as not to break the roots. After filling the 20 × 25 × 3 cm transparent glass tray with water, the washed roots and spread were added for scanning (Epson Perfection V700, Seiko Epson Crop., Suwa, Japan). For the morphological analysis of root morphology, the WinRHIZO program (Regent, CA/WinRHIZO Pro. LA2400 Scanner) was used to analyze the morphology of root development in the container. The scanned image is the total root length, total projected root area, total root surface area, total root volume, and average root diameter of the *Q. myrsinifolia.* In addition, RCD (root collar diameter) was measured, and 4 grades of RCD (less than 0.5, 0.5–1.0, 1.0–1.5, 1.5–2.0 mm) were marked at 0.5 mm intervals. Containers in which plants were planted were not fertilized during the experiment period.

### 2.5. Growth Characteristics and Quality Index of Seedlings

Six specimens were collected at random per treatment, and height and root diameter were measured as indicators of the growth characteristics of seedlings. Then, the collected seedlings were completely dried at 75 °C for 72 h in a drying oven (CT-FDO 42, Anyang, Korea), and the leaves, stems, and roots were divided into parts and weighed with an electronic balance (HS114B, Seoul, Korea). The following formula was used to obtain H/D rate, T/R rate, dry weight ratio, and root dry weight ratio by using the measured index (Table 2).

**Table 2.** Formulas for measuring shoot and root growth of *Q. myrsinifolia* Blume seeds.

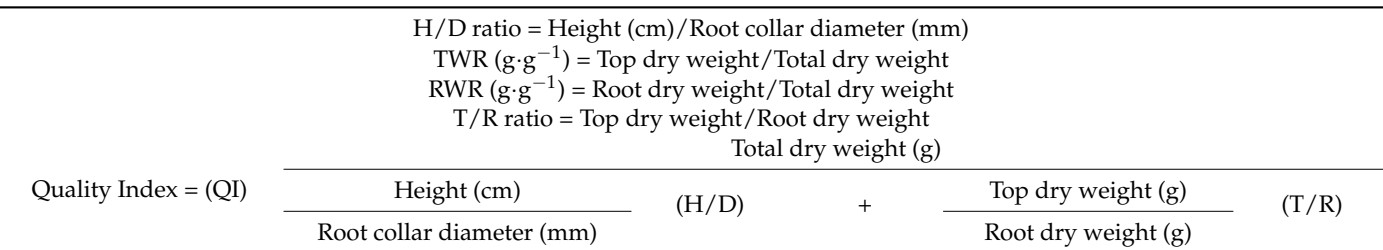

The Quality Index (QI) of the seedlings was based on the evaluation of the health of seedlings (H/D rate), the balance between the growth of top and root at the ground (T/R rate), etc., for planting and growth after healthy planting [16]. In a way designed to predict performance, the quality index of seedlings was obtained by using the measured values of height, root diameter, and dry matter production (Table 2).

### 2.6. Statistical Analysis

The germination rate and growth characteristics of the species used in the experiment were analyzed by collecting 40 specimens for each treatment group in 3 repetitions. The collected data were subjected to statistical analysis by using SPSS software version 25. One-way analysis of variance (ANOVA) was conducted, and means were compared using Duncan's multiple range test (DMRT) at a 0.05 level of probability. Values were represented as mean ± standard deviation (SD).

## 3. Results

### 3.1. Relationship between Seed Moisture Content and Germination Rate

*Q. myrsinifolia* seeds showed decreasing moisture content and germination rate over time (Figure 1). The moisture content began to decrease rapidly as drying started. The moisture content, from 36% at the time of seed harvest, reached 30% after 5 days, 10% after 10 days, 8% after 14 days, and 5% after 21 days. On the other hand, the germination rate decreased rapidly as drying started. Seeds that showed a 97% germination rate immediately after harvesting decreased to 37% after 5 days, showed 10% after 14 days and did not germinate after that. Therefore, it was found that the seed moisture content and

the germination rate showed a high correlation. When the moisture content decreased, the germination rate also decreased. The critical drying period of the seeds was 21 days at 25 °C.

### 3.2. Germination Rate by Removal of the Seed Coat

Germination of *Q. myrsinifolia* seeds took place between 10 and 14 days after sowing, and the germination patterns were different for each seed. In seedlings, the hypocotyl grew longer in the early stage, and the cotyledon grew later (Figure 2A–G). In the case of stored seeds, after 2 weeks of sowing, the leaves were not dark but showed a complete plant shape, but some seeds had slow germination (Figure 2H). Eight weeks after sowing, sprout growth was promoted, and the chlorophyll content of the leaves was higher than before (Figure 2H). After 5 months, it could be seen that the plant had grown regardless of the germination rate (Figure 2I).

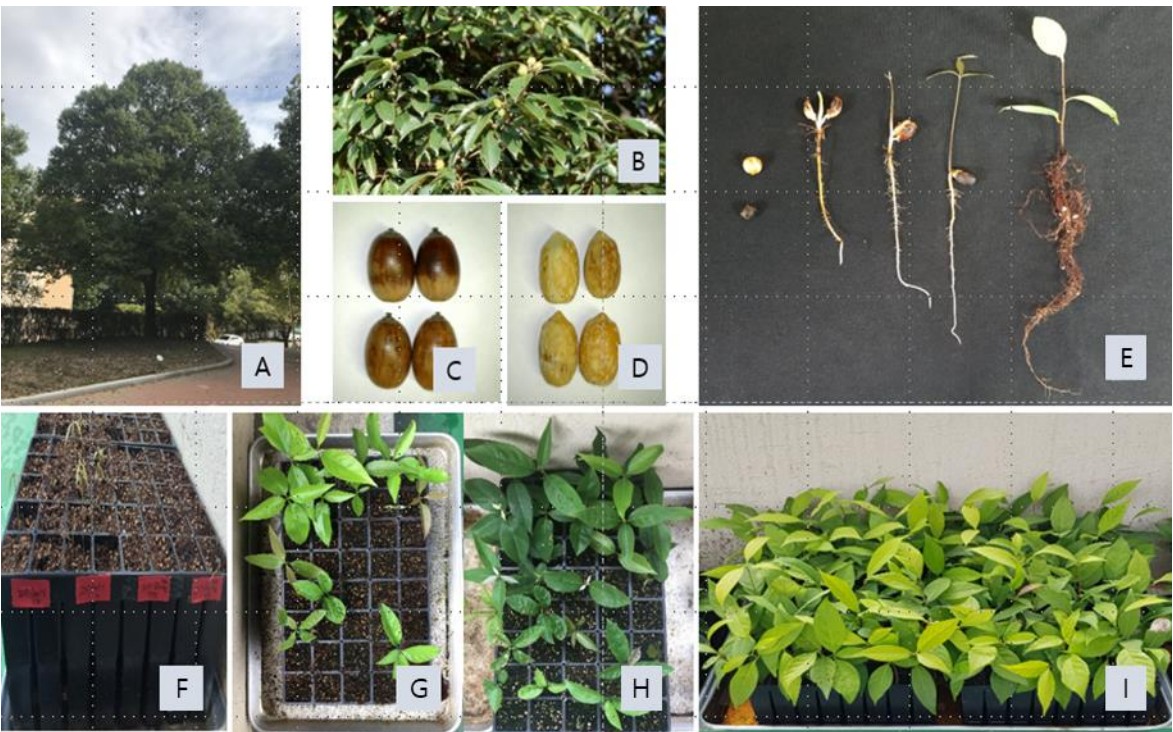

**Figure 2.** Germination of *Q. myrsinifolia* seeds and further growth of seedlings. (**A**) *Q. myrsinifolia* from which the seeds for this study were collected; (**B**) *Q. myrsinifolia* seeds; (**C**) intact seed; (**D**) seeds with seed coat removed; (**E**) seed germination pattern; (**F**) germination of seed with seed coat removed (2 weeks after sowing); (**G**) germination of seeds with the seed coat removed (4 weeks after sowing); (**H**) germination of seeds with the seed coat removed (8 weeks after sowing); (**I**) 5 months after sowing seedlings.

*Q. myrsinifolia* seeds have been shown to have different germination rates and germination characteristics depending on the storage period and the presence of a seed coat (Table 3). The GP of the seeds with seed coat was 26.67–74%, the MGT was 12.83–30.83, the GR was 0.37–0.6, and the GV was 0.37–0.9. On the other hand, when the seed coat is removed, GP is 51–88%, MGT is 14.4–21.93, GR is 0.8–1.1, and GV is 0.81–2. The GP was very high in the seed from which the seed coat was removed compared to the intact seed. For GP, the germination rate was higher than 30% for fall seeding, 13% from seeds stored for 1 year, and more than 42% from seeds stored for 3 years. MGT was different depending on whether or not the seed coat was removed. In the year in which the seeds were collected, the MGT of the seed with the seed coat was 22.07 days, but the MGT of the seed with the seed coat removed was 21.93 days.

**Table 3.** Germination pattern according to the storage years of seeds and the presence or absence of seed coats.

| Storage Periods | Treatment | GP (%) | MGT | GR | GV |
|---|---|---|---|---|---|
| 0 | Control | 29.33 ± 5.03 [b*] | 22.07 ± 1.81 [b] | 0.56 ± 0.14 [c] | 0.40 ± 0.15 [c] |
| | Remove seed coat | 51.33 ± 6.43 [a] | 21.93 ± 2.77 [b] | 1.04 ± 0.35 [a] | 0.81 ± 0.22 [bc] |
| 1 year | Control | 74.01 ± 5.29 [b] | 12.83 ± 2.02 [a] | 0.60 ± 0.26 [b] | 0.90 ± 0.26 [b] |
| | Remove seed coat | 87.33 ± 6.11 [a] | 14.40 ± 2.51 [a] | 1.10 ± 0.36 [a] | 2.01 ± 0.26 [a] |
| 3 years | Control | 26.67 ± 8.33 [b] | 30.83 ± 5.01 [c] | 0.37 ± 0.31 [c] | 0.37 ± 0.21 [c] |
| | Remove seed coat | 88.01 ± 9.17 [a] | 20.40 ± 2.51 [b] | 0.80 ± 0.36 [ab] | 1.22 ± 0.28 [b] |

GP: Germination Percent; MGT: Mean Germination time; GR: Germination Rate; GV: Germination Value). * The data are presented as means ± SD of independent tests performed three times. A different letter for each treatment shows significant difference using Duncan's Multiple Range Test (DMRT) comparison ($p < 0.05$).

The MGT of the seeds stored for 1 year with the seed coat removed was 14.4 days, and for the seeds without the seed coat removed, this was 12.83 days. Among the seeds stored for 2 years, the MGT of the seed without the seed coat was 20.4 days, and the MGT of the seed with the seed coat was 30.83 days.

GR was also different depending on whether or not the seed coat was removed. The GR was higher in the seeds from which the seed coat was removed than the seeds without the seed coat. In the case of seeds collected in the current year, when the seed coat was removed, the GR was 1.04, and when the seed coat was not removed, the GR was 0.56. The GR of seeds stored for 1 year was 0.6 when the seed coat was not removed and 1.1 when the seed coat was removed. In the case of seeds stored for 2 years, the non-removal of the seed coat was 0.37 and 0.8 when the seed coat was removed.

GV was also different depending on whether or not the seed coat was removed. For the seeds collected in the current year, the GV was 0.81 when the seed coat was removed and 0.4 when the seed coat was not removed. The GV of seeds stored for 1 year was 0.9 when the seed coat was not removed and was 2 when the seed coat was removed. In the case of seeds stored for 2 years, the non-removal of the seed coat was 0.37 and 1.22 when the seed coat was removed.

In the case of lateral roots, seedlings germinated from seeds with seed coats were 4.08 cm long, and seedlings from which seed coats were removed were 4.82 cm. The number of leaves was 2.40 in seedlings from seeds from which the seed coat was removed, which was much higher than 1.1 in seedlings germinated from intact seeds. Therefore, when the seed coat was removed, the RCD, shoot length, root length, growth and number of leaves increased, which is effective for leaf growth (Table 4).

**Table 4.** Growth characteristics of seedlings germinated from intact seeds and seeds from which the seed coat has been removed.

| Treatment | RCD (mm) | Shoot Length (cm) | Root Length (cm) | Rootlet Length (cm) | Leaf Number | Leaf Length (cm) |
|---|---|---|---|---|---|---|
| Intact seed | 2.44 ± 0.64 [b*] | 8.4 ± 2.86 [b] | 2.77 ± 0.59 [b] | 4.08 ± 2.71 [a] | 1.1 ± 1.66 [b] | 1.9 ± 2.51 [b] |
| Seed coat removal seed | 3.38 ± 0.64 [a] | 11.44 ± 2.06 [a] | 3.42 ± 0.50 [a] | 4.82 ± 3.43 [a] | 2.4 ± 1.44 [a] | 4.81 ± 2.12 [a] |

* The data are presented as means ± SD of independent tests performed three times. A different letter for each treatment shows significant difference using Duncan's Multiple Range Test (DMRT) comparison ($p < 0.05$).

### 3.3. Growth Characteristics and QI of Seedlings by Potting Soil Types

The number of leaves, leaf length, height, and RCD of seedlings of *Q. myrsinifolia* were investigated by potting soil type (Table 5). Factors involved in the growth of seedlings showed different patterns depending on the type of potting soil.

**Table 5.** Height, RCD and H/D ratio of seedlings according to potting soil type.

| Potting Soil * | Height (cm) | RCD (mm) | H/D Ratio | Leaf Number | Leaf Length (cm) |
|---|---|---|---|---|---|
| CPS | 10.34 ± 2.05 | 2.56 ± 1.23 [ab] | 4.78 ± 2.03 | 4.75 ± 1.39 [a**] | 7.48 ± 1.67 [a] |
| Sandy soil | 10.23 ± 2.27 | 1.84 ± 0.29 [b] | 5.74 ± 1.82 | 2.38 ± 1.19 [b] | 4.96 ± 2.33 [b] |
| CPSS | 11.23 ± 2.38 | 2.94 ± 1.14 [a] | 4.25 ± 1.40 | 4 ± 1.69 [a] | 7.46 ± 1.07 [a] |

* All investigations were performed after growing the 1-year-old seedlings in a greenhouse for 4 weeks. CPS is a commercially available potting soil, and CPSS is a mixture of CPS and sandy soil. ** The data are presented as means ± SD of independent tests performed three times. A different letter for each treatment shows significant difference using Duncan's Multiple Range Test (DMRT) comparison ($p < 0.05$).

The height of seedlings was the highest at 11.23 cm in CPS and sand mixed potting soil (CPSS), followed by 10.34 cm in CPS, and the smallest at 10.23 cm in potting soil using sand. RCD was the largest at 2.94 mm in CPSS potting soil and the smallest at 1.84 mm in potting soil using sand. The H/D ratio was the highest at 5.74 in sandy soil, and the lowest at 4.25 in CPSS potting soil. The number of leaves was the highest at 4.75 in CPS, CPSS mixed soil and the lowest at 2.38 in sandy soil. Leaf length was the longest at 7.48 cm in CPS and 4.96 cm in sandy soil. As a result of examining the growth of seedlings according to the type of soil, the H/D ratio was in the range of 4.25 to 5.74.

Image analysis was performed using the WinRHIZO program for the root shape of each seedling planted in 24 cavities (Figure 3). A three-day irrigation cycle promoted the root development of 1-year-old and 3-year-old seedlings in all potting soils. However, in the case of 1-year-old seedlings, root development was weak, and it was difficult to infer the root shape according to the treatment of soil by image. It can be observed that the root growth was vigorous in CPS and CPSS potting soil from 1-year-old and 3-year-old seedlings (Figure 3).

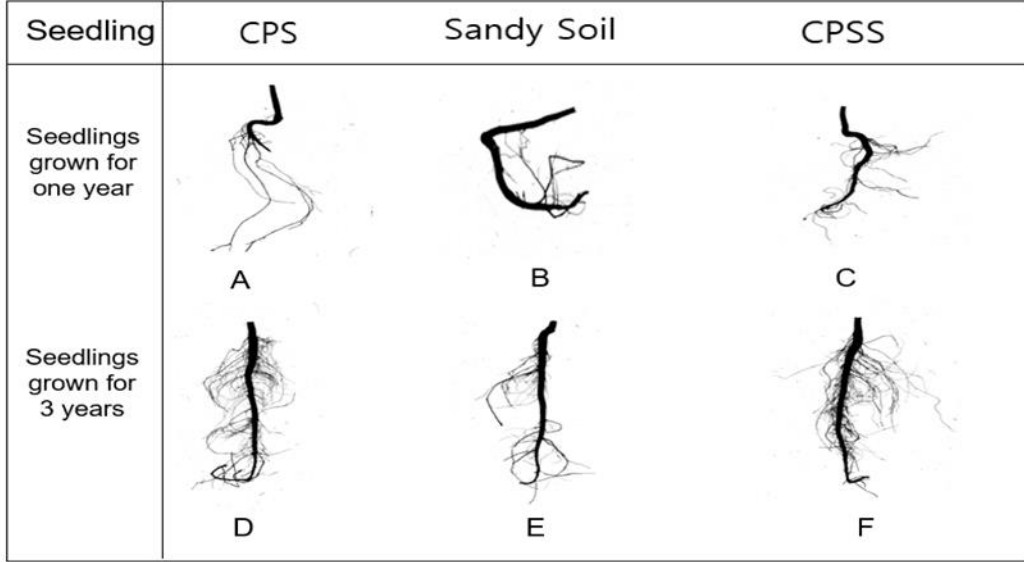

**Figure 3.** Growth pattern of seedling roots of 1 and 3 years according to potting soil types. CPS is a commercially available potting soil, and CPSS is a mixture of CPS and sandy soil. (**A**) 1 year seedling root in CPS, (**B**) 1year seedling root in Sandy soil, (**C**) 1 year seedling root in CPSS, (**D**) 3 years seedling root in CPS, (**E**) 3 years seedling root in Sandy soil, (**F**) 3 years seedling root in CPSS.

Depending on the potting soil, the total root growth of the seedlings had a significant effect (Table 6). In 1-year-old seedlings, the total root length was the longest at 80.47 cm in CPS, 74.62 cm in mixed soil, and 37.19 cm in sandy soil. Projected root area and root surface area also tended to be similar to root length. The average root diameter was the

longest at 1.03 mm in sandy soil, 0.68 mm in mixed soil, and 0.6 mm in biomass. The root volume was also the highest in the sandy soil at 0.3 cm.

**Table 6.** Root growth of 1-year-old and 3-year-old seedlings according to potting soil types.

| Treatments * | | Total Root Length (cm) | Projected Root Area (cm$^2$) | Root Surface Area (cm$^2$) | Total Root Volume (cm$^3$) | Average Diameter (mm) |
|---|---|---|---|---|---|---|
| 1 year grown seedlings | CPS | 80.47 ± 18.48 [a]** | 4.88 ± 0.83 | 15.33 ± 2.61 | 0.23 ± 0.03 | 0.6 ± 0.05 [b] |
| | Sandy soil | 37.19 ± 8.81 [b] | 3.79 ± 0.62 | 11.91 ± 1.96 | 0.3 ± 0.03 | 1.03 ± 0.07 [a] |
| | CPSS | 74.62 ± 14.75 [a] | 5.16 ± 1.72 | 16.22 ± 3.00 | 0.28 ± 0.09 | 0.68 ± 0.11 [b] |
| 3 years grown seedlings | CPS | 443.51 ± 136.69 [a] | 23.52 ± 6.03 [a] | 73.75 ± 18.9 [a] | 0.98 ± 0.2 [a] | 0.55 ± 0.04 [b] |
| | Sandy soil | 122.71 ± 25.34 [b] | 8.92 ± 1.68 [b] | 28 ± 3.01 [b] | 0.51 ± 0.10 [b] | 0.72 ± 0.08 [a] |
| | CPSS | 195.78 ± 96.83 [b] | 12.81 ± 5.29 [b] | 40.18 ± 16.63 [b] | 0.66 ± 0.22 [ab] | 0.69 ± 0.07 [ab] |

* All investigations were performed after growing the 1-year-old seedlings in a greenhouse for 4 weeks. CPS is a commercially available potting soil, and CPSS is a mixture of CPS and sandy soil. ** The data are presented as means ± SD of independent tests performed three times. A different letter for each treatment shows a significant difference using Duncan's Multiple Range Test (DMRT) comparison ($p < 0.05$).

In the case of 3-year-old seedlings, the root growth of seedlings showed a significant difference depending on the potting soil types. The total root length of the seedlings was the longest at 443.51 cm in CPS, followed by 195.78 cm in mixed soil and 122.71 cm in sandy soil. The projected area, surface area, and root surface area also showed similar trends. The average root diameter was longest in sandy soil at 0.72 mm, followed by 0.69 mm in mixed soil and 0.55 mm in CPS.

The type of potting soil affected the fresh and dry weight of roots, T/R rate, and seedling quality index (Table 7). The total dry weight of seedlings was high in the order of top and bottom. Both fresh and dry soils were the best in CPS, followed by mixed soil and sand. The T/R ratio was the highest at 1.26 in CPS and the lowest at 1.05 in sandy soil. In addition, the seedling quality index was the highest at 0.2665 in mixed soil, indicating good quality, and the lowest in sandy soil.

**Table 7.** Fresh weight and dry weight, T/R rate and seedling quality index of seedlings on potting soil.

| Treatment * | Fresh Weight(g) | | | Dry Weight(g) | | | T/R Ratio | Quality Index *** |
|---|---|---|---|---|---|---|---|---|
| | Top | Root | Total | Top | Root | Total | | |
| CPS | 1.48 ± 0.44 [a]** | 1.21 ± 0.55 | 2.69 ± 0.99 | 0.59 ± 0.18 [a] | 0.58 ± 0.29 | 1.17 ± 0.47 | 1.26 ± 0.63 | 0.266 ± 0.15 |
| Sandy soil | 0.77 ± 0.33 [b] | 0.67 ± 0.33 | 1.44 ± 0.66 | 0.33 ± 0.14 [b] | 0.33 ± 0.17 | 0.65 ± 0.31 | 1.05 ± 0.20 | 0.1665 ± 0.14 |
| CPSS | 1.44 ± 0.60 [a] | 1.14 ± 0.57 | 2.58 ± 1.16 | 0.61 ± 0.20 [a] | 0.53 ± 0.26 | 1.15 ± 0.46 | 1.21 ± 0.18 | 0.2665 ± 0.15 |

Each datapoint was measured after 1 month. The data are presented as means ± SD of independent tests performed three times. Means with different letters are significantly different at a = 0.05 by Duncan's multiple range test. * CPS is a commercially available potting soil, and CPSS is a mixture of CPS and sandy soil. ** The data are presented as means ± SD of independent tests performed three times. A different letter for each treatment shows significant difference using Duncan's Multiple Range Test (DMRT) comparison ($p < 0.05$). *** The quality index of seedlings was obtained using the method of [16], and measured values of height, root diameter, and dried product yield were used.

### 3.4. Growth Characteristics and Quality Index of Seedlings by Irrigation

Periodic irrigation treatment was found to affect the growth of *Q. myrsinifolia* seedlings (Table 8). It was found that daily irrigation is good for seedling growth. The growth of the seedlings irrigated every day for 5 months was 10.78 cm; for the seedlings irrigated every 3 days, this was 10.34 cm; and for the seedlings irrigated every 7 days, this was 10.04 cm.

**Table 8.** Height and RCD and H/D ratio of seedlings on irrigation.

| Irrigation Period (Day) | Height (cm) | RCD (mm) | H/D Ratio | Leaf Number | Leaf Length (cm) |
|---|---|---|---|---|---|
| 1 | 10.78 ± 1.85 * | 1.49 ± 0.84 [b] | 11.3 ± 9.05 [a] | 4.17 ± 1.59 | 8.51 ± 1.27 [a] |
| 3 | 10.34 ± 2.05 | 2.56 ± 1.24 [a] | 4.78 ± 2.03 [b] | 4.75 ± 1.39 | 7.48 ± 1.67 [ab] |
| 7 | 10.04 ± 2.64 | 2.08 ± 0.71 [ab] | 5.08 ± 1.42 [b] | 3.75 ± 1.48 | 7.21 ± 1.01 [b] |

Irrigation experiments were performed for 5 months, and irrigation was performed at intervals of 1, 3, and 7 days. * The data are presented as means ± SD of independent tests performed three times. A different letter for each treatment shows significant difference using Duncan's Multiple Range Test (DMRT) comparison ($p < 0.05$).

Additionally, RCD was most significant when the irrigation cycle was 3 days, but there was no significant difference according to the irrigation cycle. The RCDs were 1.49 mm for daily irrigation, 2.56 mm for 3-day irrigation and 2.08 mm for 7-day irrigation.

The H/D ratio was 4.78–11.3. In particular, the H/D ratio was the highest at 11.3 during daily irrigation and the lowest at 4.78 mm during the 3-day irrigation cycle.

The number of leaves also affected the irrigation cycle. The number of leaves was highest in the 3-day cycle, and the number of leaves was significantly lower in the 7-day interval. The leaf length was the longest when irrigation was carried out at daily irrigation, and the 7-day intervals had the most petite leaf length.

According to the irrigation days, the root growth of 1-year-old and 3-year-old seedlings was different (Figure 4). In the case of the 1-year-old seedlings, root development was inadequate, but in the case of the 3-year-old seedlings, the development of the main root and the lateral root occurred relatively vigorously. The root development of the 1-year-old seedlings showed strong root growth when irrigated every day or once every 7 days, and multi-stranded lateral roots developed. Additionally, 3-year-old seedlings showed the best root development when irrigated every 3 days.

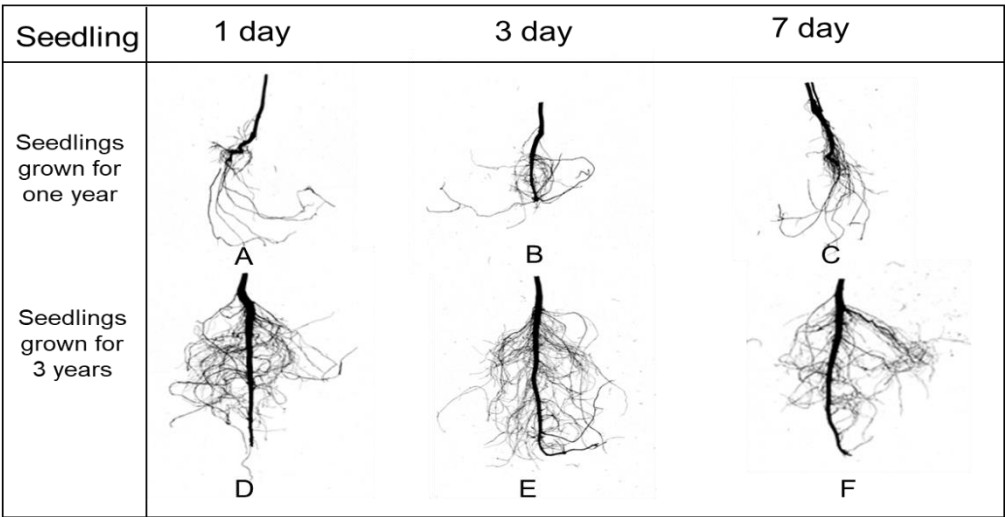

**Figure 4.** Growth pattern of seedling roots for 1 and 3 years according to irrigation period. (**A**) 1 year root in 1 day cycle, (**B**) 1year root in 3 day cycle, (**C**) 1 year root in 7 day cycle, (**D**) 3 year root in 1 day cycle, (**E**) 3 year root in 3 day cycle, (**F**) 3 year root in 7 day cycle.

The WinRHIZO program analyzed seedling root development concerning the irrigation cycle (Table 9). According to the irrigation cycle, the root growth of 1-year-old and 3-year-old seedlings showed a significant difference. The total root length of the 1-year-old seedlings was the longest at 103.51 cm when irrigated every 7 days, followed by 102.65 cm at daily irrigation and 80.47 cm every 3 days. Projection area, surface area, and root volume also showed similar trends with the total root length. The average root diameter was

0.601 mm, the longest when irrigated for 7 days, and the lowest during daily irrigation (Table 9).

**Table 9.** Growth of 1- and 3-year-old seedlings according to irrigation.

| Seedling | Irrigation | Total Root Length (cm) | Projected Root Area (cm$^2$) | Root Surface Area (cm$^2$) | Total Root Volume (cm$^3$) | Average Diameter (mm) |
|---|---|---|---|---|---|---|
| 1 year grown seedling | 1 day | 102.65 ± 13.46 [a] | 6.05 ± 1.83 [a] | 19.02 ± 2.64 [ab] | 0.28 ± 0.03 [ab] | 0.59 ± 0.15 |
| | 3 days | 80.47 ± 18.48 [b] | 4.88 ± 0.83 [b] | 15.33 ± 2.61 [b] | 0.23 ± 0.03 [b] | 0.61 ± 0.05 |
| | 7 days | 103.51 ± 1.71 [a] | 6.35 ± 0.20 [a] | 19.98 ± 0.61 [a] | 0.3 ± 0.02 [a] | 0.61 ± 0.03 |
| 3 years grown seedling | 1 day | 409.52 ± 19.23 * | 21.85 ± 1.05 | 68.57 ± 2.00 | 0.91 ± 0.11 | 0.54 ± 0.07 |
| | 3 days | 443.51 ± 136.69 | 23.52 ± 6.03 | 73.74 ± 18.90 | 0.98 ± 0.20 | 0.55 ± 0.04 |
| | 7 days | 366.93 ± 23.18 | 19.54 ± 3.51 | 61.19 ± 8.05 | 0.78 ± 0.02 | 0.53 ± 0.04 |

Irrigation experiments were performed for 5 months, and irrigation was performed at intervals of 1, 3, and 7 days. * The data are presented as means ± SD of independent tests performed three times. A different letter for each treatment shows significant difference using Duncan's Multiple Range Test (DMRT) comparison ($p < 0.05$).

In the case of the 3-year-old seedlings, the root growth pattern was slightly different from that of 1-year-old seedlings. Total root length was the longest at 443 cm when irrigated every 3 days and the shortest at 366 cm when irrigated every 7 days. Projection area, surface area, and root volume also showed similar trends with the total root length. The average root diameter was most significant during the 3-day irrigation cycle and the smallest during the 7-day (Table 9).

The total dry weight of the seedlings according to the irrigation cycle was investigated (Table 10). The total dry weight of seedlings also differed according to the irrigation cycle. The total weight of the seedlings was found to be heavy in both fresh and dry weight with 3-day irrigation, followed by 7-day irrigation, and daily irrigation was the lightest. It was found that the weight of the aboveground part was slightly heavier than that of the underground part.

**Table 10.** Fresh weight and dry weight, T/R rate and seedling quality index of seedlings on irrigation.

| Treatment ** (Day) | Fresh Weight(g) | | | Dry Weight(g) | | | T/R Ratio | Quality Index |
|---|---|---|---|---|---|---|---|---|
| | Top | Root | Total | Top | Root | Total | | |
| 1 | 0.94 ± 0.11 [b]* | 0.74 ± 0.47 | 1.67 ± 0.44 [b] | 0.38 ± 0.04 [b] | 0.32 ± 0.20 | 0.7 ± 0.18 [b] | 1.65 ± 0.99 | 0.137 ± 0.09 |
| 3 | 1.48 ± 0.44 [a] | 1.21 ± 0.55 | 2.69 ± 0.99 [a] | 0.59 ± 0.18 [a] | 0.58 ± 0.29 | 1.17 ± 0.47 [a] | 1.26 ± 0.63 | 0.266 ± 0.15 |
| 7 | 1.09 ± 0.11 [b] | 0.76 ± 0.42 | 1.83 ± 0.51 [ab] | 0.45 ± 0.06 [ab] | 0.34 ± 0.20 | 0.79 ± 0.26 [ab] | 1.61 ± 0.73 | 0.166 ± 0.08 |

Irrigation experiments were performed for 5 months, and irrigation was performed at intervals of 1, 3, and 7 days. * The data are presented as means ± SD of independent tests performed three times. A different letter for each treatment shows significant difference using Duncan's Multiple Range Test (DMRT) comparison ($p < 0.05$). ** The quality index of seedlings was obtained using the method of [16], and measured values of height, root diameter, and dried product yield were used.

The T/R ratio was the highest at 1.65 for seedlings irrigated every day and lowest at 1.26 for seedlings irrigated with a 3-day cycle.

The quality index of seedlings was the highest at 0.266 for seedlings irrigated with a 3-day cycle and 0.137 for seedlings irrigated every day.

## 4. Discussion

Seeds that can be safely dried to a moisture content between 6 and 10% and safely dried and stored successfully at low temperatures are termed 'orthodox seeds', and seeds that cannot be dried to this level without losing viability are termed recalcitrant seeds [17]. Temperate recalcitrant species include *Quercus* species. In general, it is known that the moisture content, germination, and amount of recalcitrant seeds decrease as drying increases. *Quercus* recalcitrant seeds are challenging to germinate when a 10–15% or less moisture content falls. This study showed that the critical germination moisture was 5%. This value

is lower than the critical (destructive) moisture content of 10–15% [18] of *Quercus nigra* acorns. Therefore, *Q. myrsinifolia* is an important tree species to maintain moisture in the storage and management of seeds.

*Quercus* species produce recalcitrant seeds, germinating at relatively high moisture content [19]. There are very few studies on why recalcitrant seeds are difficult to germinate. Studies to date have shown that recalcitrant seeds do not accumulate dehydrin, abscisic acid (ABA) or soluble sugars during seed maturation [20].

After harvesting the seeds, the experiment was conducted after drying them indoors for 7 days. Since the species in this study is a recalcitrant plant, it is necessary to consider the effect of indoor drying for 7 days on the germination rate. Further studies are needed on 7 days of room drying to determine seedling quality and further growth.

The removal of the seed coat had a positive effect on the germination of *Q. myrsinifolia*. Seed moisture affects the quality and further growth of the seedling. The GP for increasing the cultivation efficiency by enabling the uniformity of germinated seedlings was 30% in the seeds without the seed coat, and the GP for the seeds with the seed coat removed was 20% higher. The seed coat may prevent germination because it interferes with water uptake and gaseous exchange and also contains chemical inhibitors, which act as the barrier against the escape of inhibitors from the embryo, modify the light reaching the embryo, and impose mechanical restraints [21,22]. With Mango cultivar Bappakkai, seed coat removal recorded maximum fresh and dry weight. It was reported that seed coat removal resulted in fresh and dry weight and increased leaf area and fresh weight, mainly due to increased seedling growth and development [23]. Kim [24] also reported that *Prunus yedoensis* germinated within 11 days due to removing the seed coat, and this removal of the seed coat was the effect of removing the germination inhibitor. Giradi et al. [25] also stated that the removal of citrus seed coat allowed the emergence of plants about 1 week earlier than intact seeds. When the seed coat was removed and sowing was performed, the root collar diameter, root diameter, stem length, root length, growth and number of leaves increased.

There are many reports on the promotion of germination rate but not many studies on seedlings' growth by removing the seed coat. In this study, the root collar diameter, root diameter, stem length, root length, growth, and the number of leaves were increased when the *Q. myrsinifolia* seed coat was removed and seeded. Nayanakantha et al. [26] reported that rubber (*Hevea brasiliensis*) seeds were removed from the seed coat and treated with nitric oxide to promote seedlings' germination rate and growth. In addition, it has been reported that the removal of the citrus seed coat promoted the germination rate and the growth of stock 25% more than the intact seed, along with the root induction and root growth [25]. Mechanical scratches, such as removal of the seed coat, have a positive effect on seedling settling and seedling growth such as seedling height and root length, seedling and root dry weight and leaf score, and enhancement in germination rate [27].

The removal of the seed coat promoted the growth of *Q. myrsinifolia* via faster absorption of nutrients than plants germinated from intact seeds. Girardi et al. [24] also reported that rapid nutrient absorption in stock production by removing the seed coat of citrus increased the root weight of the stock.

The irrigation cycle had a significant effect on the growth of seedlings, RCD, H/D ratio, and leaf growth. The easiest factor in investigating the growth characteristics of seedlings is height and root diameter, which are the primary factors for selecting the standard of seedlings planted in afforestation [28].

As a result of examining RCD, rootstock growth and dry matter production of container seedlings of pine trees irrigated at intervals of 1, 2, 3, 5, 7, 10, and 15 days, the best growth response was found to be highest at intervals of 1 day of irrigation [29]. Additionally, the shorter the irrigation cycle, the better the growth of container seedlings of pine trees, but the total root length was higher at 3 days than at 2 days or 1 day.

Daily irrigation is suitable for the seedling length and root collar diameter, and the longer the irrigation cycle, the lower the growth tendency [7]. The height and root collar diameter of yellow poplars were also the best in the once/day irrigation treatments [30].

However, in the case of this study, it is judged that the growth and quality of seedlings according to the irrigation cycle differed from species to species, as the 3-day irrigation cycle showed the best quality of seedlings.

The irrigation cycle has been shown to influence root development [30]. Roots are usually responsible for fixing plants in one place, absorbing moisture and inorganic nutrients from the soil, and storing carbohydrates. The root length to diameter distribution ratio can be considered an important characteristic when describing or comparing root systems growing in soil. These roots are an important indicator of the quality of seedlings, and weak root growth promotes soil growth by increasing the uptake of water and nutrients by seedlings and affects the overall yield of seedlings [27]. Therefore, the effect of seedling root development on the number of irrigation days was visually confirmed using the WinRHIZO program. It is complicated to investigate the factors affecting the root growth of seedlings, such as irrigation and potting soil in nurseries. It is particularly difficult because field studies of root system dynamics require continuous and non-destructive measurements [31]. The image analysis system can morphologically and rapidly analyze various plant roots. The commercial package WinRHIZO is a root analysis system that can measure roots' shape, area, and arrangement. Himmelbauer et al. [32] investigated the average root diameter, root surface area, and root arrangement using the roots of wheat and barley grown in the field and confirmed that the reproducibility was very high. Although this root imaging system has been studied for herbaceous plants, the efficiency of container nurseries can be further improved using this imaging device for nurseries of woody plants.

The irrigation cycle altered RCD, root weight, T/R ratio and seedling quality. The T/R ratio is the weight of the upper part that increases the weight of the root, which is responsible for water absorption of the seedling and is a means of measuring the balance between the aboveground part and the underground part [33]. When the height is standard, the lower the T/R ratio, the better the seedling quality [34]. Looking at the change in T/R ratio according to the irrigation cycle, the *Q. myrsinifolia* seedlings irrigated once every 3 days had the lowest value of 1.03. Additionally, as a result of the seedling quality index survey, the seedlings irrigated once every 3 days had the highest quality index of 2.57. The seedling quality index was equal to the total dry weight.

These results suggest that root development of *Q. myrsinifolia* container seedlings with irrigation is closely related to growth in height, root diameter, and total dry weight. In other words, when the root growth is active, the ability to absorb water and nutrients is developed, and the soil growth, overall seedling production, and dry weight for each tissue increase. Therefore, root growth is a good indicator for evaluating the quality of seedlings. The amount of irrigation every 3 days is less than when irrigating every day because of early growth. It is judged that the growth of the roots is insufficient. It is necessary to monitor the growth status of the transplanted seedlings in the future.

The potting soil affected the growth of *Q. myrsinifolia* seedlings. The best potting soil for seedling length was CPSS potting soil, and CPSS potting soil was also good for the T/R ratio and QI. The reason the mixed soil was good is thought to be because it maintained appropriate nutrients such as N, P, and K and proper soil porosity. Most potting soil is developed for horticulture, so it may not be suitable for a nursery of forest species. Commercial potting soil contains all essential minerals and nutrients necessary for plant growth and development. It is known to give vitality to seed germination and growth [35]. The CPS used in this study was a mixture of 20% coco peat, 59.26% peat moss, 20% perlite, 0.632% dolomite, 0.0008% wetting agent, and 0.1% fertilizer. However, this study found that it is better to use a mixture of sandy soil than to use all CPS for *Q. myrsinifolia* nurseries. In other words, it means that physical factors such as minerals, nutrients, and pore space in the potting soil are important in *Q. myrsinifolia* nurseries. Supporting this is the growth pattern on sandy soils. Sandy loam does not contain sufficient minerals, so growth is poor, and the quality of seedlings is poor. Mohan et al. [36] reported that the growth of Neem tree seedlings was active on a mixture of farm manure with 30 ppm nitrogen and 20 ppm phosphorus, sandy and clay soils.

This study focused on improving the germination rate and subsequent growth promotion effect by removing the seed coat and improving the seedling efficiency through irrigation and potting soil. *Q. myrsinifolia*, an evergreen oak tree species known to be difficult to nurse, should produce healthy and high-quality seedlings. In this study, it is necessary to pay attention to the H/D ratio and quality index for the health of seedlings.

According to the "How to Implement Seedling Projects" after the revision on 1 July 2014, in the case of broad-leaved trees (1-year-old seedling), H/D rate values of 7.0 or less are reported as healthy seedlings [32]. The H/D ratio is the value obtained by dividing the height by the root collar diameter and is an index representing the health of the seedlings produced [28,32,34]. The higher the H/D ratio, the thinner and weaker the seedlings, and the lower the H/D ratio, the thicker and stronger the seedlings [34]. In the experiment by irrigation treatment, the H/D ratio of *Q. myrsinifolia* seedlings (1-year-old seedlings) was found to be in the range of 3.42 to 4.36, which is considered a healthy seedling [37].

QI is a quality index of seedlings obtained by considering the H/D ratio and T/R ratio. Generally, a high value of QI is recognized as a healthy seedling [28,38]. In this study, the QI of *Q. myrsinifolia* seedlings was 0.124–0.257 in the irrigation cycle test and 0.149–0.262 in potting soil. Cha et al. [29] reported that the QI values of pine seedlings were 0.071, 0.067, and 0.065 when irrigated at intervals of 1, 2, and 3 days, respectively.

Figure 5 summarizes a series of processes for continuously supplying high-quality seedlings of *Q. myrsinifolia*. To proliferate excellent *Q. myrsinifolia* seedlings, the seeds should be collected between September and December, dried in the shade for 7 days, and then stored at a low temperature of 4 degrees. There will be no problem with germination even if the seeds are stored for 3 years. In the oak family, the haunting seed phenomenon is severe, and to continuously supply seedlings, it is essential to preserve the seeds. In addition, to promote the germination and growth of seeds, it is important to remove the seed coat before sowing. The potting soil used for container nurseries is considered to increase the nursery's efficiency by mixing general horticultural potting soil and sand in a 1:1 ratio. Additionally, irrigation every 3 days is appropriate for the growth of actual seedlings. Through this series of processes, high-quality seedlings can be mass-produced in a short time. Therefore, it is judged that the conditions found in this study can nurture excellent seedlings.

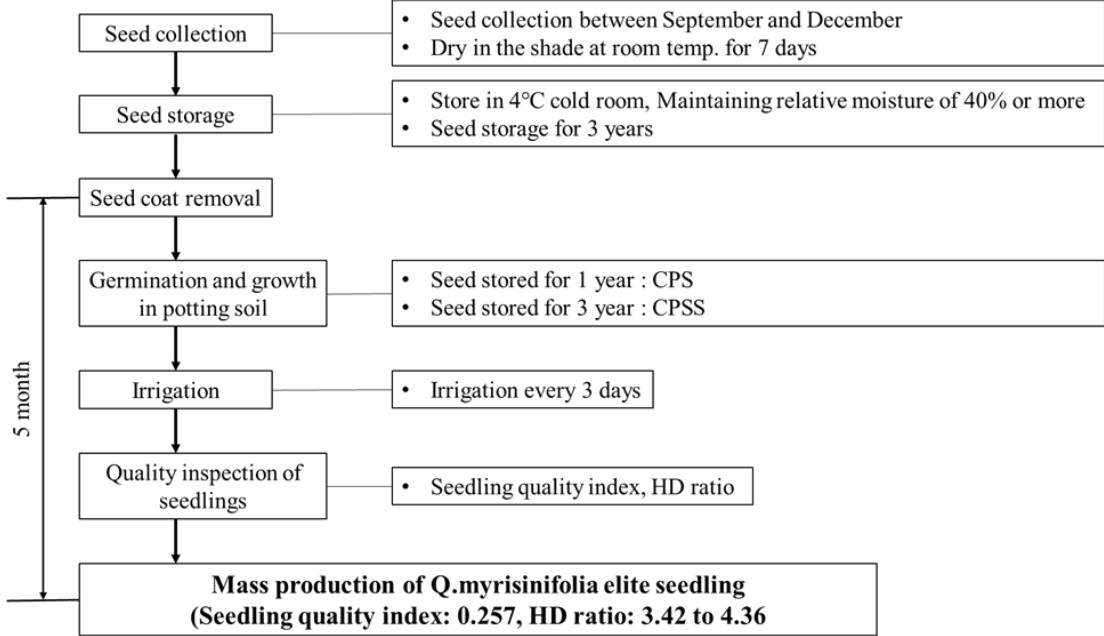

**Figure 5.** Schematic diagram for seedling propagation of *Q. myrsinifolia* Blume. CPS: commercially available potting soil; CPSS: CPS and sand mixed soil (CPSS: sand, 1:1, *v/v*).

## 5. Conclusions

In Northeast Asia, including Korea, fine dust and carbon dioxide emissions rapidly increase due to rapid industrialization, threatening human health. *Q. myrsinifolia* is an evergreen oak tree species that absorbs carbon dioxide and fine dust and can be used as expensive wood, agricultural and industrial materials. However, asexual reproduction, such as through the use of cuttings, is complicated and has the characteristics of recalcitrant seeds, so the importance of seedling propagation is increasing. This study promoted germination and growth of germinated seedlings by removing the seed coat of seeds stored for 3 years. As a result, the germination rate was higher in the seed with the seed coat removed than in the seed without treatment. The germination rate of GP was higher than 30% during culturing, 13% from seeds stored for one year, and more than 42% from seeds grown for three years. In addition, when the seed coat was removed, RCD, shoot length, growth and leaf number increased compared to intact seeds, leading to increased root length and growth of lateral roots. These results will improve the germination rate of *Q. myrsinifolia* seeds and produce high-quality seedlings quickly. Additionally, the type of potting soil had a significant effect on the growth of seedlings. For the growth of *Q. myrsinifolia* seedlings, it was found that adding sand to commercially available potting soil improves the growth, T/R rate and seedling quality index of the aboveground and underground parts. This result can reduce the cost of purchasing potting soil, which is an economic burden on the nursery industry. Irrigation also greatly affected the seedling, and the three-day irrigation period increased the number of leaves and improved root growth. This result laid the groundwork for reducing water consumption, another important economic limiting factor in the nursery industry. Overall, nursery practices established in this study will produce high-quality *Q. myrsinifolia* seedlings with a high seedling quality index. In addition, it is judged that high-quality seedlings can be produced in a short time if applied to the nursery of evergreen oaks.

**Author Contributions:** Conceptualization, M.-S.C. and S.-H.Y.; methodology, E.-J.C., K.-B.P. and D.-H.K.; resources, E.-J.J. and S.-H.Y.; writing—review and editing, E.-J.J., S.-H.Y. and D.-J.P.; visualization, S.-H.Y.; funding acquisition, M.-S.C. All authors have read and agreed to the published version of the manuscript.

**Funding:** This study was carried out with the support of "Forest BioResource Collection, Conservation and Characteristic Evaluation" of the National Forest Seed and Variety Center and "The Forestry Science and Technology Research and Development Project" (Forest Convergence Specialist Training Project, Project No. 2020186A00-2022-AA02) in 2021.

**Data Availability Statement:** Not applicable.

**Acknowledgments:** We would like to thank the Warm Temperate and Subtropical Forest Research Center of the National Academy of Forest Sciences for cooperating with the collection of seeds.

**Conflicts of Interest:** The authors declare no conflict of interest.

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
