# Peer review of "Germination and Growth Characteristics of Quercus myrsinifolia Blume Seedlings According to Seed Coat Removal, Type of Potting Soil and Irrigation Cycle"

_forests, doi:10.3390/f13060938_

Round 1

Reviewer 1 Report

After reviewing the manuscript, I think most of the points in my previous review still stand. The manuscript needs to be further reviewed because part of it is difficult to understand and ambiguous due to the choice of wording. The second major problem is that the Methods were not reviewed and are still written with disregard for the statistical results.

Other problems:

The growing media composition need to be explained in the methods because there is a confounding factor in the soil treatment. The commercial media seems to have supplemental nutrients while the other soil mixtures do not.

The statistical analysis need to be described in more detail.

Minor problems:

“Q. myrsinifolia Blume” Blume only needs to be mentioned in the first reference of the species.

Line 79: distance to what?

Line 114: I’m not sure if the acorns were removed with the tools from the trees or the seed coat was removed with said tools.

Line 158, 179: 10L per plant, or tray?

Line 238: in which treatment?

Line 313: this belongs to MM

Line 314: compared to what treatment?

Line 513+: this belongs to MM

Line 525: unclear

Line 540: The best in terms of what?

There are more similar issues throughout the manuscript, IT needs a thorough review.

I have to conclude again that “I think the topic and experimental work are certainly of relevance, results are interesting and have a great potential for a good manuscript. However, the manuscript at the moment is unfinished.”

Author Response

Q & C(Question & Comments) : After reviewing the manuscript, I think most of the points in my previous review still stand. The manuscript needs to be further reviewed because part of it is difficult to understand and ambiguous due to the choice of wording. The second major problem is that the Methods were not reviewed and are still written with disregard for the statistical results.

Q & C(Question & Comments) : The growing media composition need to be explained in the methods because there is a confounding factor in the soil treatment. The commercial media seems to have supplemental nutrients while the other soil mixtures do not. The statistical analysis need to be described in more detail.

  • Regarding the composition of the soil used in the experiment, a detailed explanation was written in item 2.4 of the materials and methods. We also made additional corrections to statistics.

Q & C(Question & Comments) : “Q. myrsinifolia Blume” Blume only needs to be mentioned in the first reference of the species.

  • We checked the entire print version and made corrections.

Q & C(Question & Comments) : Line 79: distance to what?

  • We checked the print version and made corrections.

Q & C(Question & Comments) : Line 114: I’m not sure if the acorns were removed with the tools from the trees or the seed coat was removed with said tools.

  • We checked the print version and made corrections.

Q & C(Question & Comments) : Line 158, 179: 10L per plant, or tray?

  • We checked the print version and made corrections.

Reviewer 2 Report

Correct one sentence in the conclusion. I commend all the repairs.

Author Response

Q & C(Question & Comments)  Correct one sentence in the conclusion. I commend all the repairs.

  • Corrected incorrect sentences.

è *As a result, the germination rate was very high in the seed with the seed coat removed compared to the seed with no treat.

Thank you for your detailed review of the manuscript. I think the reviewer's comments have improved the quality of the paper. Thanks again to the reviewers. I hope you are in good health.

Reviewer 3 Report

In my opinion Authors considerably improved the manuscript. They added additional experiments to strength the value of the manuscript. I reccomend to accept this manuscript after minor revision related to small edtorial mistakes and several suggestions which authors ahould consider:

Abstract: conider deletion of abbreviations which are not used for the second time in abstract.

Keywords: consider addition of "seed storage"

Introduction: consider merging lines 46-49 and then 81-85 in one pargaraph by transfering lines 46-49 before lines 81-85

the common style for writing temperature in English is using the space between the value and degree sign for example "4 °C" not "4°C", consider this change throughout the manuscript

l. 253 delete "(GP)" because it is not the abbreviation of germination rate and GP was introduced before in Material and Methods secion

 condier whether "3- year grown sedlings" should be changed to plural form  "3- years grown sedlings", number three indicates plural form (for exapmle, lines 315, 320, 394, 408); all tables contains phrase "3 years grown seedlings"

l.380 something is missing at the end of the sentence

l. 387 the dot is necassary at the end of this sentece, accept it

l. 441 "temperature" seems to be wrongly used because values are given as percentage

consider the transfer of lines  452-253 before 449-451 and merge in one paragraph

consider merging lines 467 -469 with the previous paragraph (again one sentence should not be separate paragraph

 l. 483-484 cosider change to "The removal of the seed coat promoted the growth of Q. myrsinifolia Blume via faster absorption of nutrients as compared to plants germinated from intact seeds."

l.521-522 one senetnce paragraph describes irrrigation and might be merged with the previous paragraph also related to irrigation

l. 531 "tree" seems to be unnecessary here

l. 571 the abbreviation QI was already introduced in l. 200

Figure 5 change "remove the seed coat" into "Seed coat removal" to keep the same style of writing in this cheme

l. 605 consider changing removing into "removal of" and  "seed stored" into "seed storage"

l. 607 "with" not "without" ?

Author Response

please check the upload file

This manuscript is a resubmission of an earlier submission. The following is a list of the peer review reports and author responses from that submission.

Round 1

Reviewer 1 Report

In forests-1380013, Choi et al. analyze the effect of seed coat removal, storage time, and irrigation on the germination and growth of Quercus myrsinifolia seedlings. They found that coat removal and irrigation had positive effects on seed germination and root growth respectively, while storage period had no effects.

I think the topic and experimental work are certainly of relevance and have a great potential for a good manuscript. However, the manuscript at the moment is unfinished.

The results section presents the effects on the average value of the variables instead of statistically significant differences. It has to be reviewed and rewritten.

Material and methods need more details, for example in the statistical analysis section.

Hypothesis need to be clearly stated in the introduction.

The abstract uses abbreviations that the reader cannot now the meaning.

There are many incomplete sentences. See first sentence of Conclusions.

Title should be reviewed.

Minor reviews

Line 6 (abstract): Seed “coat” removal

M&M: Do all deeds come from one single tree?

Line105-109: Needs a better explanation of the abbreviation system.

Line 125: Beed soils need further explanation.

Line 184: You have a Results and Discussion section and then a Discussion again.

Line 279 belongs to M&M

Line 502: Review

Reviewer 2 Report

I think you need to change the title of the paper because the genus Quercus have pericarp (outer, thick and solid part) and the seed coat or testa (brown, thinner inner part). You just removed the pericarp and the seed coat in the work and not just the seed coat . Along with the Latin name of the species, the name of the author is always written, for example, in your case it is Blume. I would add another keyword and separate the terms irrigation and bed soil. You have a lot of connected words in the text, so you need to proofread the entire paper. I have suggested in the comments some better synonyms for expert terms so consider them. You did an analysis of the quality of seedlings according to the author Dickson, so this should be included in the bibliography and the Dickson quality index (DQI) should be written everywhere in the text, tables and figures. All numbers in the tables should be placed in the same decimal place for easier visibility. In the tables, the labels a, b or ab (result of statistic analysis) are missing in some columns, and it is necessary to add and explain what two asterisks (**) mean to you ?. Pictures number 3 and number 5 are quite opaque to you, so I suggest that you put two tables instead of these two pictures. In the pictures there are large oscillations on the abscissa so it is very opaque and difficult to understand. The discussion is well written but check the exact surnames of the authors you cite in working with those in the bibliography. All praise in picture number 6. The conclusions are a little too general for you, I would like to remove some sentences and put yours from the research results. I suggest that future research on the morphology, growth and development of roots, mycorrhiza, diseases and pests on the root (nematodes, etc.) be done by a non-destructive method using devices such as CI-600 In-Situ Root Imager or newer model CI-602 Narrow Gauge Root Imager . It is a proven scientific instrument. 

Reviewer 3 Report

This study focused on improving the germination rate and subsequent growth promotion effect by removing the seed coat in Q. myrsinifolia acorns. After testing soil medium and irrigation Authors provided a scheme for practical use in tree nursery.

Major points:

  1. Q. myrsinifolia seeds display recalcitrant behavior, therefore they are extremely sensitive to desiccation a and very fast lose viability after drying below more or less 30% of moisture content (MC) or even 40% in some oak species. MC of acorns determines their storage capacity and further germination and seedling emergence. Approximately 40% MC is recommended for safe storage of some oak species since they are desiccation sensitive (recalcitrant).

Authors did not measure MC after collection, what is more Authors further dried acorns for a week without any control of water loss. Drying rates are unknown. Nothing is written about it in Introduction. What is the limit for acorns drying in this species?

Data of MC after drying and MC at storage was ignored what is a major mistake in the experimental plan. Therefore the MC of seeds stored for 1 and 3 years and not the storage time might be the driver of observed difference. None protocol for recalcitrant seeds exist without monitoring of MC.

Suggestion to dry recalcitrant seeds for one week without monitoring MC is not acceptable. MC in acorns at harvest differs depending on environmental conditions and at harvest time can be lower than recommended when recommended MC for storage is ~40%.

  1. Another methodological mistake in experimental plan refers to comparison of seedlings:

1-year and 3-year-old compared plants were:

1-0 seedlings grown  for one year from 1-year-old seeds

3-0 seedlings grown for 3 year from 3-years-old seeds

Presented two variants are not good model to compare. Authors should  monitor MC and compare 1yo seedlings from seeds stored for 1 and 3 years and compare  3yo seedlings from seeds stored for 1 and 3 years, because germination and seedling performance depends on seed quality resulting from their MC and storage time, both affecting aging rates

  1. Seed recalcitrance, which makes seed storage very complicated, is totally omitted in the text in Introduction sand Discussion.
  2. Figure 1 should be placed before Table 3 in relation to their citation in the text.
  3. The topic is not novel, seed-coat removal considerably shortened the period from sowing to germination in many species including Grevillea cultivars, Cucumis melo, Calophyllum brasiliense, Prunus yedoensis, Ardisia crenata and many more.
  4. Table 3 should contain “seeds” term not seedlings because refers to germination ofseeds stored for 1 and 3 years. It is not clear whether germination in the current year is related to seeds collected in 2017 or 2019 year?
  5. Remove descriptions containing many numeric values which are in the table . For example “The GP of the seeds with seed coat was 26.67~74%, the MGT was 12.83~30.83, the GR was 0.37~0.6, and the GV was 0.37~0.9.” Focus on most important data and contrasting results.

Delete from descriptions sentences describing data which are not statistically different. For example: 

“The growth of the seedlings irrigated every 1 day for 5 months was 10.78 cm, the seedlings irrigated every 3 days was 10.34 cm, and the seedlings irrigated every 7 days was 10.04 cm”

8. Why and how leaf length was measured? Leaf area is a most common parameter for leaves.

  1. Whole paragraph (lines 43-46; 83-86; 87-91) without a reference is not suitable in introduction. One sentence long paragraph are also not suitable (lines 504-505).

Except the major fallacy at the beginning on experiment (effect of MC in acorns), the idea to test soil media and irrigation has high practical meaning. However, the observed differences in seedling response to various soil media and irrigation might be as well the effect of  acorns MC at storage what determines their quality, aging rates, etc.

Minor points:
line 21 extra space
line 26 explain abbreviation
line 29 remove repetitions or a add commas to separate phrases, add a space where needed
line 35 consider adding the common name of the analyzed species to the keywords
line 77 "the" not "The" 

lines 96-97 relocate toAcknowledgements
lines 104, 105  "-year stored seeds"
line 126 "L" earlier "ml" unify the format of liter throughout the manuscript
line 128 "seeds" not "germination"
lines 131-2 rewrite, " germination values (GV)" is repeated
Table 1, check whether "i" in formulas should be written as subscript format
consider changing the structure of Table 1, because writing formulas as bulletpoints in a table are unusual
line 144 CBS was earlier explained
line 145 correct or explain CBSS

line 190, 193 a space is missing

line 406, rewrite “The RCD affected the RCD”

line 575, rewrite “In this study, the forest ecosystem is changing according to climate change.”